# A Seabed Terrain Feature Extraction Transformer for the Super-Resolution of the Digital Bathymetric Model

**Wuxu Cai** [1], **Yanxiong Liu** [1,2,*], **Yilan Chen** [1,2], **Zhipeng Dong** [1], **Hanxiao Yuan** [1] and **Ningning Li** [3]

1   The First Institute of Oceanography, Ministry of Natural Resources, Qingdao 266061, China; caiwuxu@fio.org.cn (W.C.); chenyilan@fio.org.cn (Y.C.); zhipengdong@fio.org.cn (Z.D.); yuanhanxiao@fio.org.cn (H.Y.)
2   The Key Laboratory of Ocean Geomatics, Ministry of Natural Resources, Qingdao 266590, China
3   State Key Laboratory of Information Engineering in Surveying, Mapping and Remote Sensing, Wuhan University, Wuhan 430079, China; liningning@whu.edu.cn
*   Correspondence: yxliu@fio.org.cn

**Abstract:** The acquisition of high-resolution (HR) digital bathymetric models (DBMs) is crucial for oceanic research activities. However, obtaining HR DBM data is challenging, which has led to the use of super-resolution (SR) methods to improve the DBM's resolution, as, unfortunately, existing interpolation methods for DBMs suffer from low precision, which limits their practicality. To address this issue, we propose a seabed terrain feature extraction transform model that combines the seabed terrain feature extraction module with the efficient transform module, focusing on the terrain characteristics of DBMs. By taking advantage of these two modules, we improved the efficient extraction of seabed terrain features both locally and globally, and as a result, we obtained a highly accurate SR reconstruction of DBM data within the study area, including the Mariana Trench in the Pacific Ocean and the adjacent sea. A comparative analysis with bicubic interpolation, SRCNN, SRGAN, and SRResNet shows that the proposed method decreases the root mean square error (RMSE) by 16%, 10%, 13%, and 12%, respectively. These experimental results confirm the high accuracy of the proposed method in terms of reconstructing HR DBMs.

**Keywords:** digital bathymetry model; seabed terrain feature; deformable convolutional layers; transformer; super-resolution

## 1. Introduction

Bathymetry provides fundamental information for all marine activities, and numerous marine scientific research endeavors rely on digital bathymetry model (DBM) data. High-resolution (HR) DBMs are commonly used for charting purposes [1]; at the same time, they play a crucial role in establishing hydrodynamic fluid models [2], constructing marine biological habitats and ecosystems [3], and other significant studies. In addition, HR DBMs are valuable for maritime search and rescue operations. For example, the search and rescue operations for Malaysia Airlines flight MH370 were severely hampered by the lack of accurate bathymetric data in the corresponding area [4]. Therefore, the acquisition of HR DBMs is of immense importance not only for scientific research but also for various social activities. However, the scarcity of HR DBMs severely limits their application and development.

HR DBMs are typically derived from ship-based acoustic surveys. On the one hand, these surveys face challenges and inefficiencies, resulting in the limited coverage of HR bathymetric data. According to statistics [5], ship-borne sonar-based HR data (≤800 m) cover only approximately 6.2% of the global seafloor. Consequently, the availability of HR bathymetric data to construct DBMs is severely limited for most of the global ocean. On the other hand, many institutions have obtained global low-resolution (LR) DBM data, including EPOTO and GEBCO. To address the shortage of HR DBMs, researchers have



employed various super-resolution (SR) methods to improve the resolution of DBMs; DBM SR aims to recover detailed topographic information from LR DBMs to generate HR DBMs. Currently, SR methods for DBMs mainly rely on traditional interpolation techniques, such as inverse distance-weighted interpolation [6], bilinear interpolation [7], bicubic spline interpolation [8], continuous curvature tensor spline interpolation [9], natural neighborhood interpolation [10], and kriging interpolation [11]. Although these interpolation methods are widely used due to their simplicity and speed, they can present certain problems, particularly when generating large-scale DBMs. One of their main drawbacks is the potential to be over-smooth, as these methods do not adequately exploit local terrain features during the interpolation process [12]. In addition to traditional interpolation methods, many researchers have employed deep learning-based methods to enhance the resolution of terrestrial digital elevation models (DEMs). These methods have the advantage of not relying on additional multisource data and instead generate HR DEMs solely through the reconstruction of LR DEMs [13]. Similarly, in situations in which data are limited, the resolution of DBMs can theoretically be improved using deep learning-based SR methods. This approach not only addresses the issue of insufficient data but also serves as a valuable source of HR data for DBM data fusion.

Current deep learning-based SR models are mainly based on convolutional neural networks (CNNs) and other networks based on CNNs, such as residual networks (ResNets), generative adversarial networks (GANs), and transformers. The pioneering CNN model for image SR is SRCNN, which consists of three convolutional layers that perform feature extraction, non-linear mapping, and image reconstruction to enhance image resolution [14]. The advancement of neural networks has highlighted their crucial role in feature extraction; however, the stacking of network layers can reduce the effectiveness of this network and lead to degradation problems [15]. To address this problem, He et al. [16] employed ResNet to extract shallow features and transmit mapping to the deep network. This identification mapping does not require any additional parameters or calculations, ensuring that the characteristics of this network do not deteriorate as the number of layers increases. In the context of DEM SR, Chen et al. [17] were the first to apply SRCNN to DEM data for SR reconstruction. Subsequently, other researchers have demonstrated the strong feature extraction capabilities and applicability of ResNet to DEM SR [18,19]. Peak signal-to-noise ratio (PSNR) evaluations of SR reconstruction accuracy are frequently performed in the CV domain. However, studies have revealed a mixed relationship between PSNR and visual quality, suggesting that a greater PSNR does not always translate into superior visual perception [20]. In order to enhance perception, Ledig et al. [20] integrated a perceptual loss function with a generative adversarial network for SR. Given the distinct characteristics of DEM data compared to natural images, Zhang and Yu [21] applied SRGAN, ESRGAN, and CEDGAN to DEM SR, and the experimental results showed that SRGAN was the most effective in terms of extracting terrain features, outperforming other GANs. This result suggests that neural networks with superior performance in the CV domain might not be equally effective when extracting topographic features. The current focus of DEM SR research is centered on fully and effectively extracting terrain features. Zhou et al. [22] introduced dual filters into ResNet to accomplish this goal, while Chen et al. [12] incorporated a spatial attention mechanism module into CNN to improve its feature extraction efficiency. Zhang et al. [23] proposed the addition of slope loss to the loss function, which significantly improved feature extraction effectiveness. Zhou et al. [24] integrated vector terrain features before outputting HR DEM, which significantly improved the accuracy of the model. To solve practical problems, researchers are now placing their research objects in hard-to-access topographic data. Jiang et al. [25] used high mountain data for the study data and constructed a new loss function in ResNet, which combined the terrain parameters of slope and curvature.

However, the development of global high-precision HR DBMs has been slow due to the challenges experienced when obtaining bathymetric data, particularly the lack of HR data. The resolution of HR DBMs is significantly lower than that of terrestrial DEMs, and

the application of SR research based on deep learning in DBMs is limited. Yutani et al. [26] utilized sparse coding and dictionary learning techniques for SR in DBMs, resulting in improved results compared to bicubic interpolation, and Hidaka et al. [27] employed deep learning for SR in DBMs for the first time and compared the SR reconstruction effects of various deep learning-based methods, including SRCNN, FSRCNN, ESPCN, SRGAN, and ESRGAN. The results were superior to bicubic interpolation but did not contribute to building a deep learning-based model for DBM terrain characteristics. Additionally, Zhang et al. [28] enhanced the resolution of the global GEBCO_2021 dataset from 15 arc seconds to 3 arc seconds using ResNet and migration learning. Although many deep learning-based SR methods utilize ResNet for DEMs, there are some general issues with this approach. First, the use of regular convolution kernels in ResNet might not be optimal for recovering different topographic features due to the irregular terrain. Second, the convolution operation does not adequately consider the influence of distant pixels. The transformer, with its self-attention mechanism, possesses strong capabilities in terms of capturing global information and long-range interactions among similar features, making it successful in various visual tasks. Some researchers [29] have employed this transformer in the computer vision domain to enhance SR accuracy. However, this transformer is currently underutilized in SR for terrain models. Zheng et al. [30] designed a transformer model for SR in DEMs via leveraging terrain self-similarity. The reconstruction effect was superior to SRCNN and SRGAN, but improvements specific to local features were not made. Additionally, SR is a low-level vision task, and using a complete transformer structure can reduce computational efficiency. Therefore, modifications to the transformer structure are necessary for its application in DBM SR.

In this paper, considering the complex and extensive nature of DBM seabed terrain features, a seabed terrain feature extraction transformer (STFET) is proposed to achieve DBM SR, combining ResNet with deformable convolutional layers and an efficient transformer. The following are this paper's key contributions:

1. This study is an early attempt to restore HR DBMs from LR DBMs using a transformer. In addition, we utilize the proposed transformer-based model, combined with ResNet and deformable convolutional layers, which ensures that STFET can capture both local and global seafloor topographic features.
2. Given the characteristics of large changes and rapid fluctuations in the DBM terrain, the traditional convolutional layers in ResNet are replaced by deformable convolutional layers, which have the ability to flexibly modify the sampling position of the convolution kernel to align with irregular features, improving the extraction of local seabed terrain features.
3. The transformer's self-attention mechanism allows long-term dependencies to be established between similar regions in the DBM, improving global terrain feature reconstruction. In addition, the matrix is broken down into smaller matrices for parameter manipulation, increasing the speed of model operations.

## 2. Methodology

The StfeT method includes the following: (1) data-processing; (2) training in the proposed model; and (3) testing and evaluation. The workflow is summarized in Figure 1. More detail is provided in other sections.

### 2.1. Network Architecture

In view of the complex terrain characteristics of DBMs, in this paper, we propose a seabed terrain feature extraction transformer model for DBM SR. This model mainly consists of three parts: shallow terrain feature extraction, deep feature extraction, including the seabed terrain feature extraction module and the efficient transformer, and sub-pixel convolution reconstruction, as shown in Figure 2. The utilization of convolutional layers offers a straightforward and efficient approach to incorporating feature maps, demonstrating a consistent performance across various computer vision tasks.

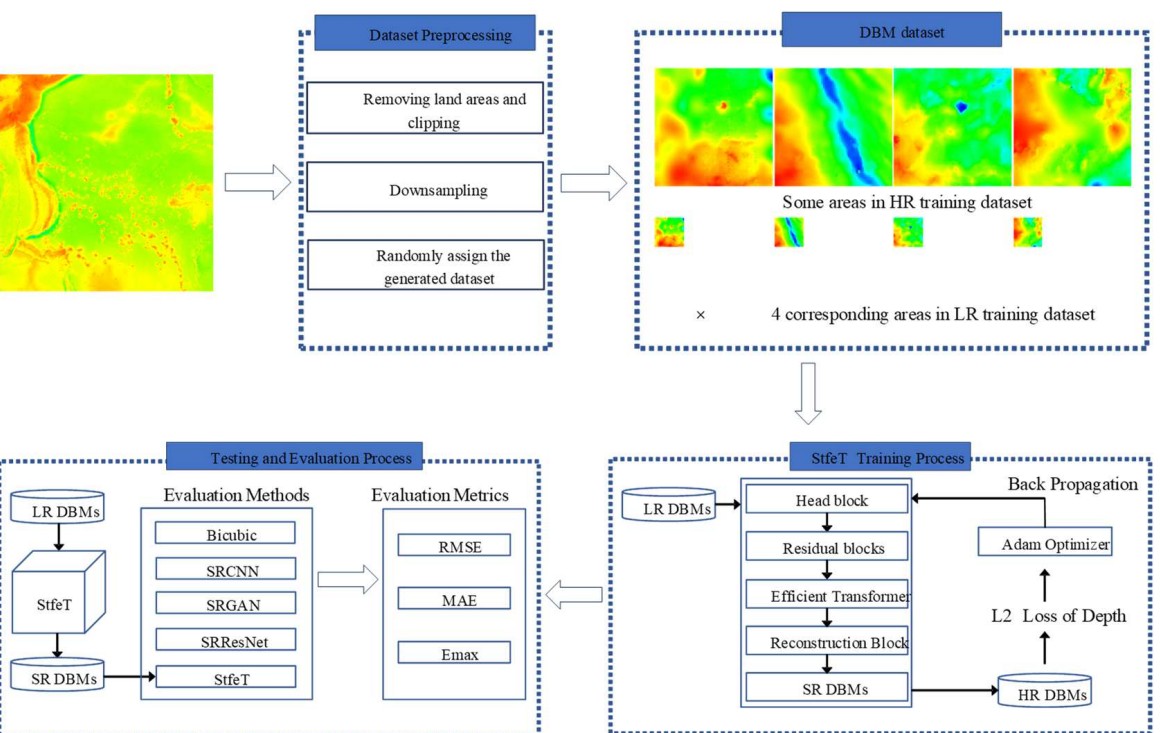

**Figure 1.** The StfeT workflow.

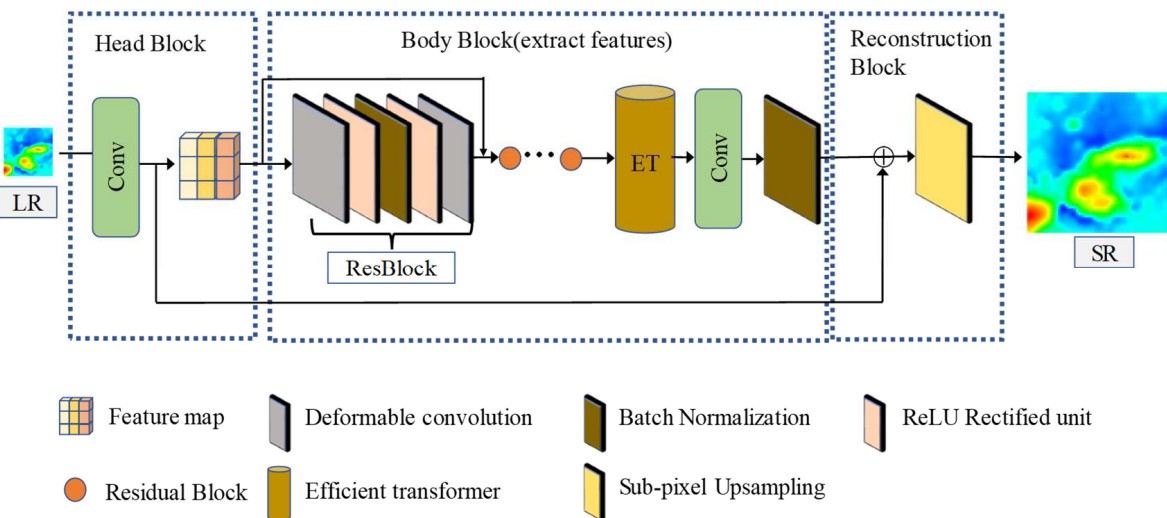

**Figure 2.** Framework of the proposed STFET.

To extract the shallow features, we employed a regular convolutional layer $f_N$ (channel = 64, kernel size = 9, stride = 1) as the first STFET module to extract the DBM feature $F_{shallow}$:

$$F_{shallow} = f_N(I_{LR}) \tag{1}$$

where $I_{LR}$ represents LR DBM. After extracting the shallow features to extract deep features, we employed the seabed terrain feature extraction module and the effective transformer module. Seabed terrain features were extracted adaptively using ResNet with deformable convolutional layers, and the intermediate features are represented by $A_1, A_2, \ldots, A_n$:

$$A_i = Res(A_{i-1}) + A_{i-1}, i = 1, 2, \ldots, n \tag{2}$$

$$A_1 = Res(F_{shallow}) + F_{shallow} \tag{3}$$

$$F_{deep} = ET(A_n) \tag{4}$$

where *Res* represents a residual block with deformable convolutional layers. Through residual learning, seabed terrain features could be effectively extracted, and the deformable convolutional layer could focus on high-frequency information, thereby enhancing the extraction of local features. *ET* represents the efficient transformer, and the feature maps after residual learning were learned using *ET*. Through the self-attention mechanism, the correlation between similar seabed terrain features was modeled, allowing the model to focus on other positions with similar seabed terrain features to extract global seabed terrain features. The deep feature $F_{deep}$ was obtained through these two modules.

After obtaining the deep seabed terrain features, the conventional convolutional layers were used to extract and transform the high-frequency features in the residual information. *BN* is batch normalization, which is then used to hasten model convergence and improve the model's robustness. Furthermore, a skip connection is used to connect the shallow topographic features with deep topographic features to obtain all topographic features $F_{all}$.

$$F_{all} = BN\left(f_N\left(F_{deep}\right) + F_{shallow}\right) \tag{5}$$

Finally, the LR feature maps were transformed into HR feature maps using sub-pixel convolution which were reconstructed to generate $I_{SR}$, which is SR DBM. During sub-pixel convolution, the HR feature maps were obtained using convolution and multichannel recombination, avoiding a large number of zero-filled areas in general deconvolution upsampling. Moreover, we used the dropout to prevent model overfitting before reconstruction.

$$I_{SR} = f_P\left(F_{deep}\right) \tag{6}$$

where $f_P$ is a sub-pixel convolutional layer. STFET uses a large number of convolutional layers to extract and transform features. The local seabed terrain features and global seabed terrain features were extracted mainly through ResNet with deformable convolutional layers and the efficient transformer. Finally, a sub-pixel convolutional layer was used to reconstruct the SR DBM.

### 2.2. Seabed Terrain Feature Extraction Module

DBM often has clear topographic undulations in local areas, generating different topographies, and the depth distribution of DBM is closely related to the topographic features of local areas. The effective extraction of seafloor topographic features can provide more accurate feature maps, thereby improving the quality of SR DBM. Very deep convolutional networks (VDSRs) [31] were the first to use ResNet for the SR domain of images, as they can utilize both deeper neural networks and mitigate the problems of disappearing or exploding gradients and network degradation. Seabed terrain features have different shapes, and the convolution kernel used in the traditional convolution layer is usually in the shape of an n × n regular grid, which makes it difficult to extract complicated seabed terrain features. To extract complex-shaped objects or terrain features, researchers have replaced conventional convolutional layers with deformable convolutional layers [32,33]. Deformable convolution adds an offset to the regular convolutional layer, which is implemented using an additional convolutional layer, as shown in Figure 3. In order to always be able to cover complicated terrain features, the deformable convolution mechanism makes sure that offsets can be learned during the learning process. Therefore, the deformable convolution layer can improve its ability to extract terrain features. In this paper, we replaced regular convolutional layers in the residual network with deformable convolutional layers, and the feature extraction process of the conventional convolutional layer and the deformable convolutional layer in the DBM is shown in Figure 4. When the conventional

convolutional layer extracts terrain features because the convolution kernel is regular, the edge terrain features are easily overlooked, whereas the deformable convolutional layer can adaptively extract seabed terrain features by learning during training.

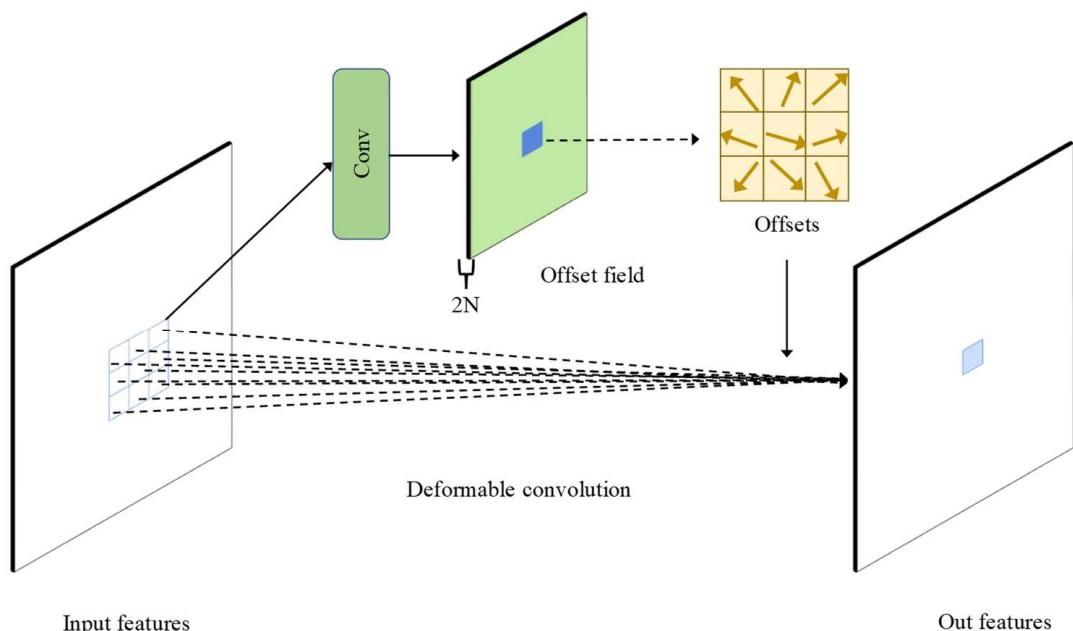

**Figure 3.** Schematic of the deformable convolution principle.

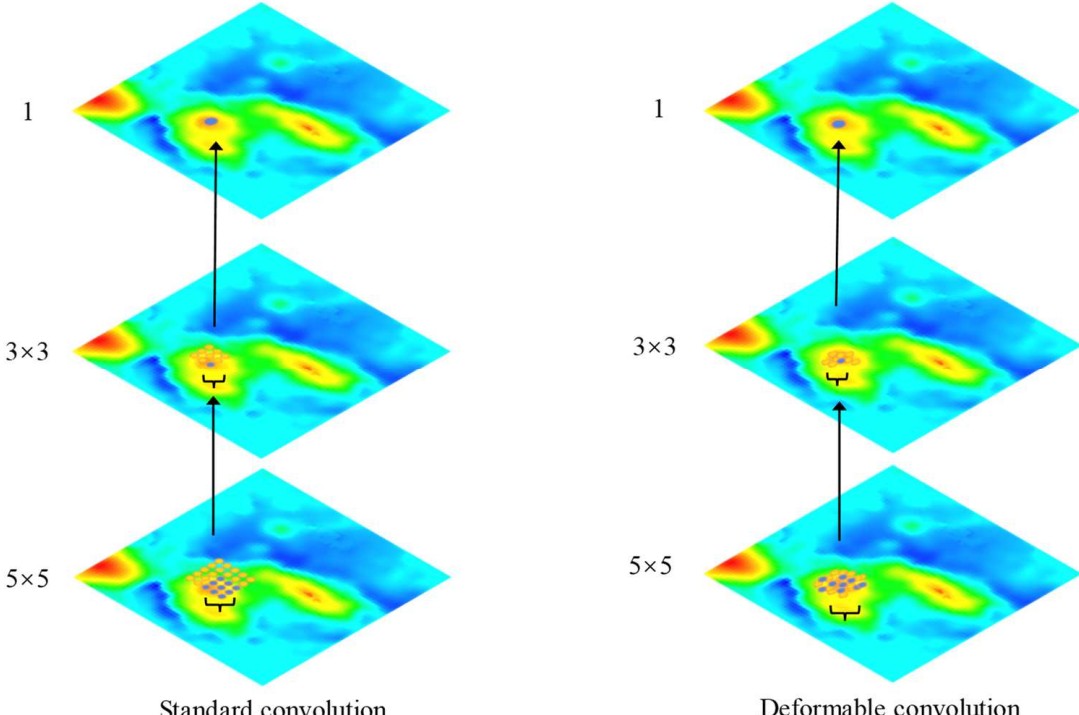

**Figure 4.** Deformable convolution and regular convolution for feature extraction on DBM.

The seabed terrain feature extraction module consists of N residual blocks. In previous studies, the higher the number of residual blocks, the higher the accuracy, but at the same time, it increases the computation time. Therefore, it is necessary to balance the reconstruction accuracy and computation time to choose an appropriate number of residual blocks. Each residual block has five layers: a ReLU activation function, two deformable

convolutional layers (channel = 64, kernel size = 3, stride = 1), and two batch normalization layers. The deformable convolutional layer is used to extract high-frequency details of the DBM to adaptively extract seabed terrain features. DBMs are different from natural images because the difference between the maximum value and the minimum value is often on the kilometer scale, and batch normalization can reduce the impact of extreme water depth. Furthermore, the ReLU activation function can enhance the non-linear relationship between layers of the neural network, which can improve the model's fitting ability. Therefore, the seabed terrain feature extraction module can effectively extract topographic features from the seafloor.

### 2.3. Efficient Transformer Module

According to the first law of geography, the closer things are, the more related they are. Therefore, in close regions, DBMs generally have a similar topography globally. The texture details of the current image block can be recovered by referring to other image blocks in single-image super-resolution (SISR), which allows similar image blocks in an image to be used as references for one another. The transformer self-attention mechanism can realize this function very well, so the transformer has a theoretical basis to recover the current area by referring to other areas to extract similar terrain features.

The structure of the efficient transformer used in this paper is shown in Figure 5. Similar to the Vision Transformer (ViT) [34], only the encoding part in the standard transformer is used, and it is composed of Multi-Layer Perceptron (MLP) and efficient multi-head self-attention (EMSA). At the same time, we used layer normalization (LN) before each module to accelerate the convergence speed of the model. Then, after each module, the residual connection was achieved on an element-by-element basis. This process can be represented using the following formula:

$$E = EMSA(LN(Un(A_n))) + Un(A_n) \tag{7}$$

$$F_{deep} = Fold(MLP(LN(E)) + E) \tag{8}$$

where $Un$ denotes the unfold function, whose purpose is to generate a local sliding block from the feature map extracted by the seabed terrain feature extraction module. $E$ represents the feature extracted using $EMSA$, and $Fold$ denotes the inverse operation of $Un$, which converts the sliding block into a feature map.

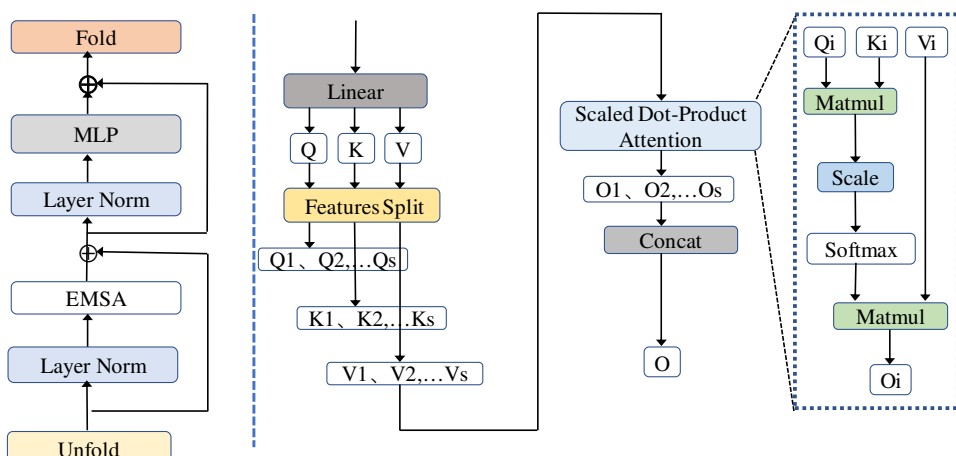

**Figure 5.** Structure of the efficient transformer and EMSA.

The EMSA structure is shown in Figure 5. First, features were embedded in query Q, key K, and value V using a linear layer. As in the transformer, we projected Q, K, and V m times, where m is the multi-head number, and the multi-head self-attention (MSA) directly computes Q, K, and V. The computations typically use a large matrix, which consumes a large amount of GPU memory and computation. Unlike MSA, the DBM matrix we used for training to extract terrain features is relatively large due to the LR of DBM and the large scale of the terrain features. Therefore, we used feature splitting to divide Q, K, and V into s parts to obtain $Q_s$, $K_s$, and $V_s$, where s is the splitting factor. Then, we use Scaled Dot Product Attention (SDPA) to perform attention to generate multiple outputs $O_1...O_s$. This process is shown on the right side of Figure 5. Finally, all outputs were concatenated to produce the overall output feature O.

## 3. Experimental Setup

### 3.1. Network Hyperparameters

In the proposed STFET, to ensure accuracy and computational efficiency, the number of residual blocks N was set to 16. The multi-head number in the efficient transformer module was set to 8, and the hidden layer size of the MLP was $mlpdim = 85$, and was obtained using the following formula:

$$mlpdim = (inchannel + outchannl) \times 2/3 \tag{9}$$

where *inchannel* and *outchannel* are the numbers of MLP input channels and output channels, respectively, which are both 64. To train the proposed STFET, we used adaptive moment estimation (Adam) with a 0.0002 learning rate, a random seed of 42, and a dropout rate of 0.2. The loss in STFET was the L2 loss MSE. For CNN-based algorithms, all the models were coded using the PyTorch framework. In addition, we used 8 GB of RAM and trained the models on a deep-learning server with an Nvidia RTX 2070S GPU. Moreover, the hyperparameters were adjusted to retrain the comparison models in the optimal way possible.

### 3.2. Study Area and Data

Unlike natural images, DBM is a grayscale image with only one channel. DBMs tend to have topographic features. In this study, the Mariana Trench in the Pacific Ocean and the surrounding waters to the east were selected as the experimental area, constituting a range of $0°-45°$N, $135°-180°$E. This area is large and rich in seafloor topography and essentially includes most of the typical seafloor topographic features, such as ridges, trenches, seamounts, sea basins, and other representative seafloor topographic features. The values of natural images are pixel values, ranging from 0 to 255. However, the elevation of our study area was between $-10{,}923$ and 3659 m. The study area is shown in Figure 6.

The DBM data used in this paper were gebco_2022_sub_ice, sourced from https://www.gebco.net/data_and_products/gridded_bathymetry_data/gebco_2022 (accessed on 10 September 2023), which has a resolution of 15 arc seconds. Since original DBM data are large, it was necessary to preprocess them to train the data. We cropped the original DBM data into $128 \times 128$ patches and removed most of the land area, and a total of 6386 DBM images were obtained as HR DBMs. To obtain paired HR DBMs and LR DBMs as datasets for training, many previous DEM SR studies were scaled 4 times for verification experiments [12,22,24]. In this study, we also used bicubic interpolation to reduce the data size 4-fold and obtain $128 \times 128$ patch DBM images with resolutions of 60 arc seconds as LR DBMs. The ratio of the training set data, verification set data, and test set data was 8:1:1 through random allocation. Before training, due to the large difference in the DBM water depth values DBMs needed to be standardized, and therefore, each DBM water depth value was standardized to $[-1,1]$. The normalization formula is expressed as follows:

$$DBM_i = 2 \times (DBM_i - H_{min})/(H_{max} - H_{min}) - 1 \tag{10}$$

where $H_{min}$ is the minimum value of DBM and $H_{max}$ is the maximum value of DBM.

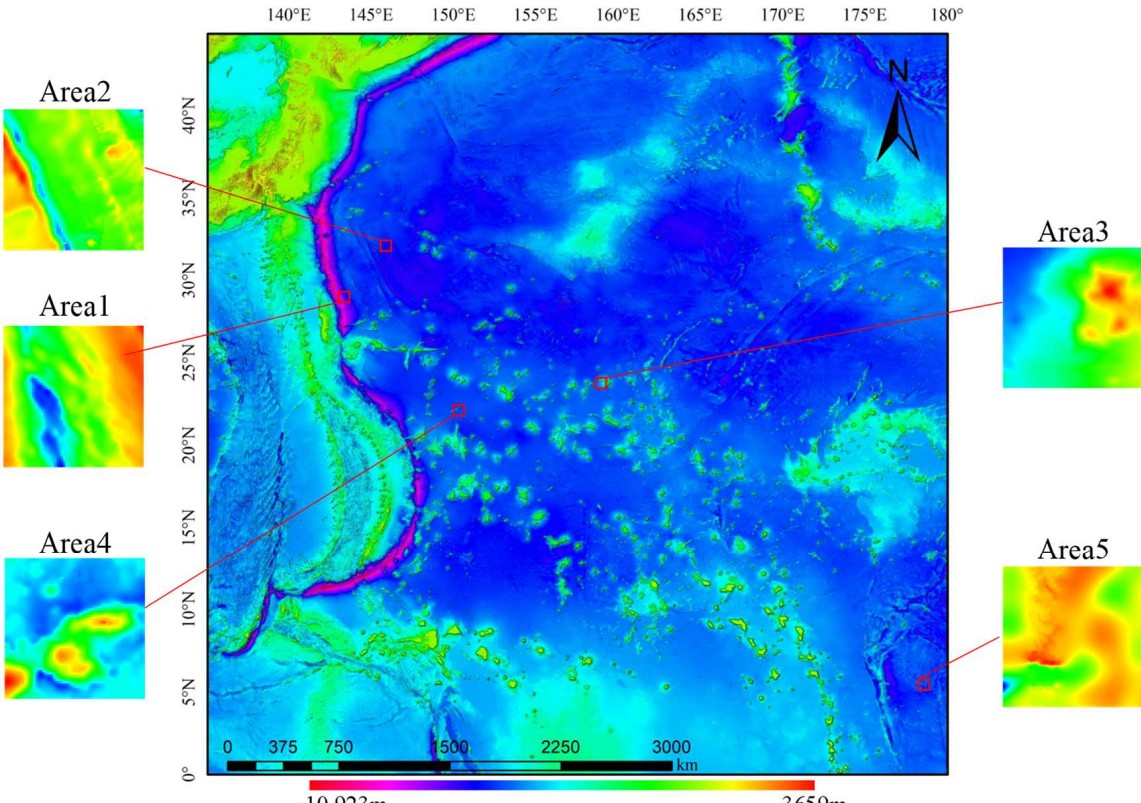

**Figure 6.** Study area.

Since the format of DBMs is usually .tif, there was only one channel, while natural images had three channels. Therefore, before inputting data into the model, we needed to set the number of input channels in the convolutional layer to 1 for training. Similarly, it was necessary to set the number of output channels of the convolutional layer to 1 before reconstruction.

### 3.3. Evaluation Methods

To test the effectiveness of STFET, we selected bicubic interpolation [35], SRCNN, SRGAN, and SRResNet (SRGAN generator) as comparison methods. Bicubic interpolation is one of the simplest and most effective interpolation methods and is often used as a baseline method for SR [21]. The first deep learning-based SR method was SRCNN. This network is simple and effective, and some scholars have proven its practicality in DEM SR. SRGAN is the first model to employ the generation confrontation network for the SR task, and it emphasizes that high accuracy does not always equate to superior visual quality using the perceptual loss function to enhance the picture. This finding has been supported by later research [36,37], and the focus of several studies has shifted from improving image accuracy to improving image data quality. Therefore, in this paper, we chose SRGAN to verify whether the perceptual loss function could acquire better visual quality in DBMs. SRResNet is the generator component of SRGAN, and it improves SRGAN's accuracy while using only the mean squared error (MSE) loss function to avoid imbalance during the training process.

DBM water depth values generally vary widely, reaching several thousand meters underwater or even tens of thousands of meters underwater. Therefore, to quantitatively compare the performance of various DBM SR methods, we used common DBM evaluation indices, the root mean squared error (RMSE) and mean absolute error (MAE), as evaluation

indices and used the maximum bathymetric error (Emax) as an evaluation index to reflect the stability of the SR methods. These evaluation indices can be calculated as follows:

$$RMSE = \sqrt{\frac{1}{N}\sum_{i=1}^{N}(h_i - \hat{h}_i)^2} \tag{11}$$

$$MAE = \frac{1}{N}\sum_{i}^{N}|h_i - \hat{h}_i| \tag{12}$$

$$E_{max} = max(abs(h_i - \hat{h}_i)) \tag{13}$$

where $N$ is the total number of pixels in a DBM image. $h_i$ and $\hat{h}_i$ represent the elevation values of the HR DBM and the corresponding SR DBM, respectively.

## 4. Results and Discussions

### 4.1. Results on DBMs

Table 1 shows the quantitative results of 638 DBMs in the test set in terms of the RMSE, MAE, and Emax indices. Among these results, those in bold highlight the best performance of each indicator, and the underlined values are the second-best results. From Table 1, we can see that, compared to other methods, STFET had the best effect on all evaluation indices. Moreover, the accuracy of all deep learning-based methods was better than bicubic interpolation, which shows that deep learning methods have an application value in DBM SR. Similar to the results for natural images [20], SRGAN achieves a lower accuracy than SRResNet based on the RMSE accuracy index.

**Table 1.** Quantitative results on the test set. Results in bold are best, and those underlined are second best.

| Method | RMSE (m) | MAE (m) | Emax (m) |
|---|---|---|---|
| Bicubic | 15.85 | 8.08 | 295.80 |
| SRCNN | <u>14.78</u> | <u>7.26</u> | <u>293.48</u> |
| SRGAN | 15.27 | 7.87 | 306.98 |
| SRResNet | 15.13 | 7.89 | 300.07 |
| STFET | **13.30** | **6.88** | **262.05** |

From Table 1, it can also be seen that SRCNN performs better than SRGAN and SRResNet; these results are different from those obtained in the CV domain. STFET uses the seabed terrain feature extraction module and effective transformer module, and SR's accuracy is significantly improved. Compared with bicubic interpolation, SRCNN, SRGAN, and SRResNet, the performance results in terms of RMSE decreased by 16%, 10%, 13%, and 12%, respectively. The test set includes complex-area and flat-area DBMs, and the STFET results are optimal, proving that this method is universal and reliable.

### 4.2. Quantitative Evaluation Results on Complex Regions

To evaluate the performance of DBM in different terrain regions, we selected five regions with more significant topographic features from the test set to conduct an accuracy comparison. These five regions contained highly undulating and complex terrain, including trenches and seamounts. The locations are shown in Figure 6. Among them, the extremum between area 2 and area 5 was small, and the terrain undulation was relatively moderate, while the topographic features of areas 1, 3, and 4 were more obvious. The original DBM for the five regions and the SR DBM reconstructed using different SR methods were combined for comparison, and the quantitative results are shown in Table 2.

**Table 2.** Quantitative results for complex regions. Results in bold are the best, and those underlined are the second best.

| Area | Method | RMSE (m) | MAE (m) | Emax (m) |
|---|---|---|---|---|
| 1 | Bicubic | 32.26 | 24.90 | <u>157.34</u> |
| | SRCNN | 31.69 | 24.72 | 201.30 |
| | SRGAN | <u>15.11</u> | <u>11.03</u> | 175.39 |
| | SRResNet | 21.40 | 16.25 | **140.28** |
| | STFET | **11.98** | **8.41** | 198.65 |
| 2 | Bicubic | 17.58 | 9.78 | 494.95 |
| | SRCNN | 17.26 | 9.73 | **483.92** |
| | SRGAN | <u>14.49</u> | <u>7.44</u> | <u>484.46</u> |
| | SRResNet | 14.69 | 7.87 | 492.02 |
| | STFET | **12.57** | **6.12** | 491.90 |
| 3 | Bicubic | 30.08 | 17.66 | 189.57 |
| | SRCNN | 29.77 | 18.04 | 196.98 |
| | SRGAN | <u>12.44</u> | <u>9.27</u> | 136.85 |
| | SRResNet | 15.22 | 10.97 | <u>126.56</u> |
| | STFET | **10.92** | **8.18** | **125.52** |
| 4 | Bicubic | 27.10 | 14.30 | 314.91 |
| | SRCNN | 26.42 | 14.63 | **295.98** |
| | SRGAN | <u>13.99</u> | <u>7.94</u> | 357.04 |
| | SRResNet | 16.69 | 9.50 | <u>310.11</u> |
| | STFET | **12.84** | **6.99** | 360.42 |
| 5 | Bicubic | 16.63 | 6.42 | 268.70 |
| | SRCNN | 15.63 | 5.05 | 279.80 |
| | SRGAN | 14.16 | 5.45 | 259.50 |
| | SRResNet | <u>12.93</u> | <u>5.27</u> | <u>207.15</u> |
| | STFET | **11.29** | **4.58** | **194.97** |

From Table 2, it can be seen that the STFET results were still optimal for complex terrain. In area 2 and area 5, all methods obtained quite accurate results, and STFET exhibited the greatest improvement. However, in area 1, area 3, and area 4, the terrain features were complex. For example, area 1 comprised a large area containing trenches with topographic features. The reconstruction effect of bicubic interpolation and SRCNN was poor, while the reconstruction effect of SRGAN and SRResNet was much better in area 1. This observation illustrates the ability of residual networks to extract features significantly. STFET has a seabed terrain feature extraction module and an efficient transformer module and, therefore, it can enhance the extraction of local terrain features and similar global terrain features so that the reconstruction results are significantly improved. Compared with bicubic interpolation, SRCNN, SRGAN and SRResNet, the accuracy in terms of the RMSE decreased by 63%, 62%, 21%, and 44%, respectively, in area 1. These results demonstrate the effectiveness of STFET in terms of extracting seafloor topographic features.

### 4.3. Visual Evaluation of Different Methods

To compare the visual quality of reconstructions using different methods, we combined HR DBMs with SR DBMs generated using different methods for the corresponding regions, and these results are shown in Figure 7. We can see that, on the whole, the SR DBM reconstructed by all methods was relatively close to the HR DBM. However, in some areas with large terrain fluctuations and obvious topographic features, subtle differences in terrain could still be observed. As shown in Figure 8, we zoomed in on the area with obvious seabed terrain features in Figure 7; the reconstruction effect of bicubic interpolation and SRCNN in areas with large terrain fluctuations was too smooth, and the visual quality was poor, which is consistent with the quantitative results in Table 2. At the same time, the proposed method had a better reconstruction effect compared to other methods, and the

result is closer to an HR DBM. Therefore, the results confirm the effectiveness of STFET in terms of both visual quality and reconstruction accuracy.

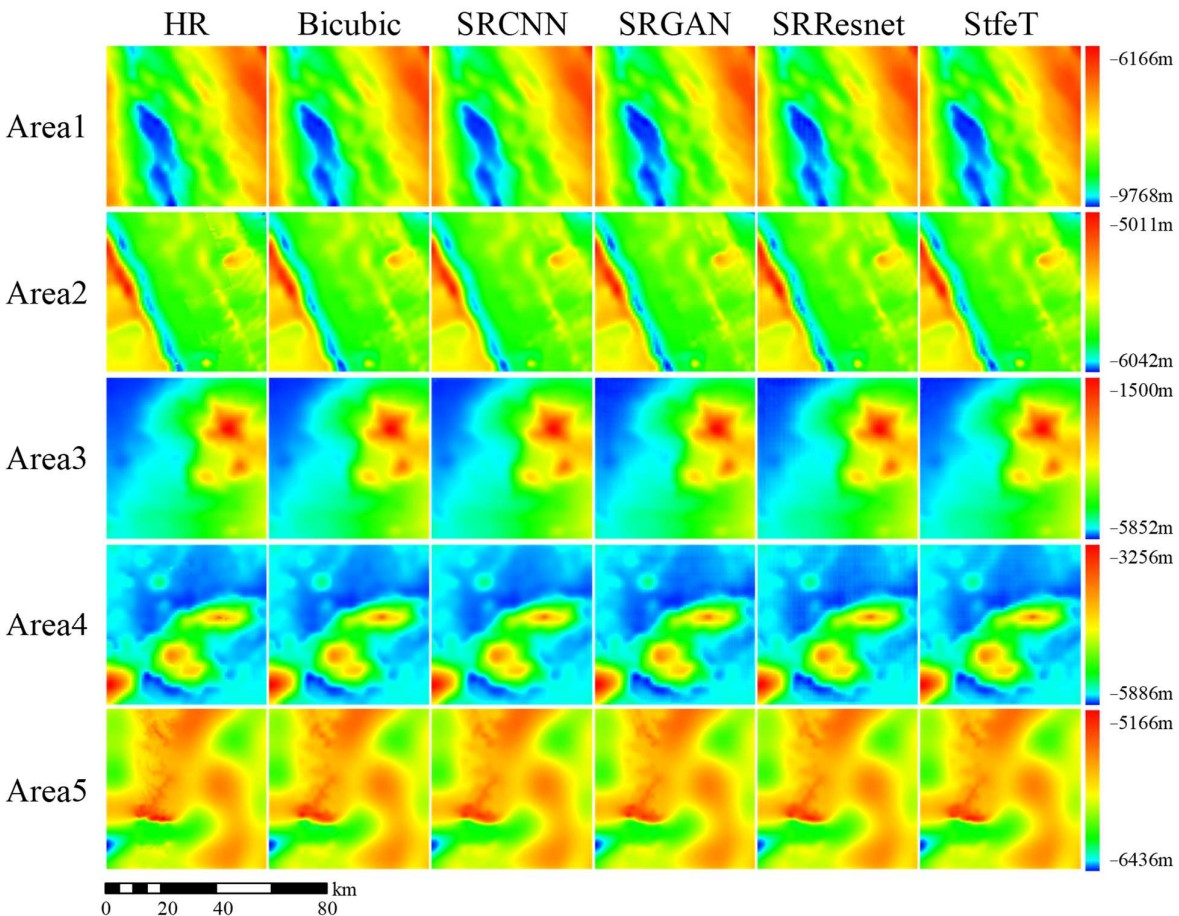

**Figure 7.** Comparison of results using different SR methods in various topographic areas.

### 4.4. Residual Evaluation of Different Methods

To visually compare the differences between the DBM and HR DBM generated using each SR method, we plotted the water depth difference between the SR DBM and HR DBM in the five districts as an error map, and the results are shown in Figure 9. The main difference in the reconstructed DBM lies in special terrain areas, such as the trench in area 1 and the seamount in area 4 in the figure, where the error is relatively large. From the error map results, we can clearly see that the error area generated using STFET was much smaller overall. The error in the local area was also greatly reduced closer to the HR DBM, which indicates that the effectiveness of the seabed terrain extraction module and the efficient transformer module was further validated. Meanwhile, similar to natural images, SRGAN outperformed SRResNet visually. However, the performance of bicubic interpolation and SRCNN was poor, and their error range was large; therefore, they were not suitable for DBM SR in complex terrain areas. As a result, the main challenge of DBM SR was to restore these areas. At the same time, it was necessary to ensure the reconstruction effect of the overall terrain.

To quantitatively analyze the error accuracy of various methods, we took 0–25 m, 25–100 m, 100–200 m, and more than 200 m as the statistical conditions and counted the number of error points generated using different methods of DBMs in Figure 9. The results are shown in Table 3. From the results, we can clearly see that the DBM reconstruction errors generated by StfeT were more concentrated at 0–25 m, and this reconstruction accuracy was better than other methods.

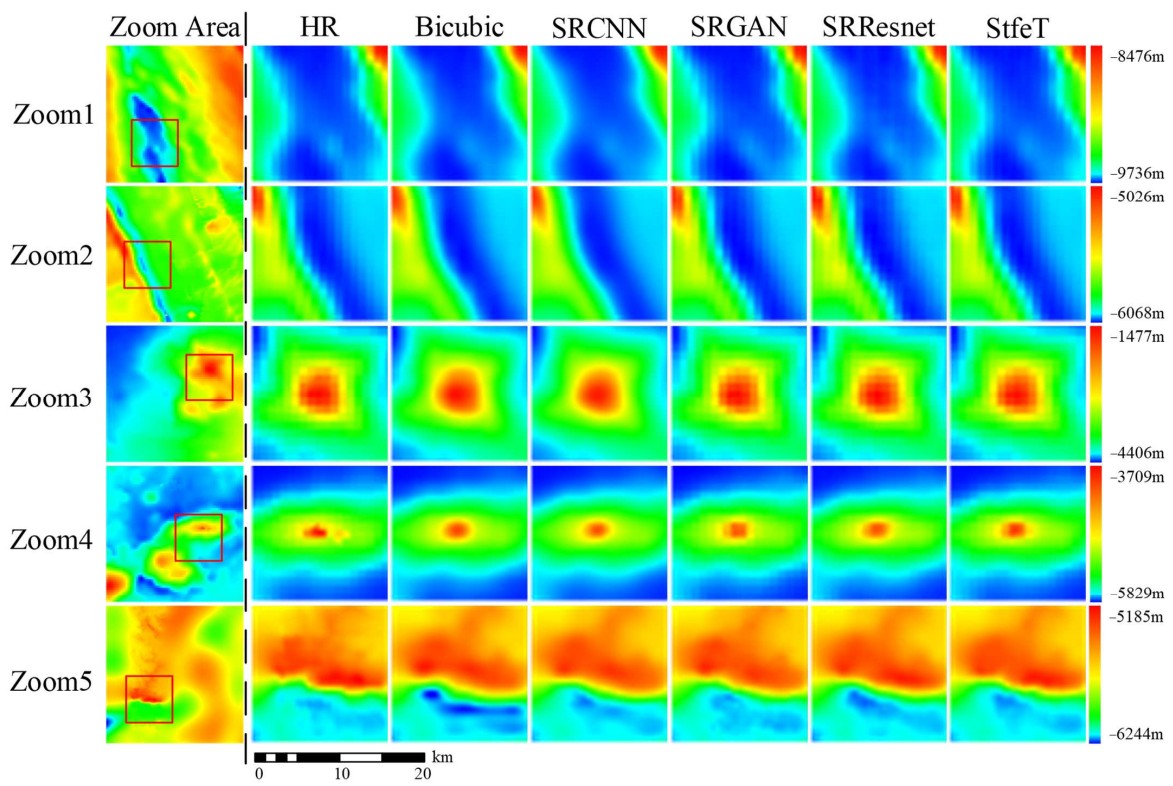

**Figure 8.** Comparison of the results after zooming in to Figure 7.

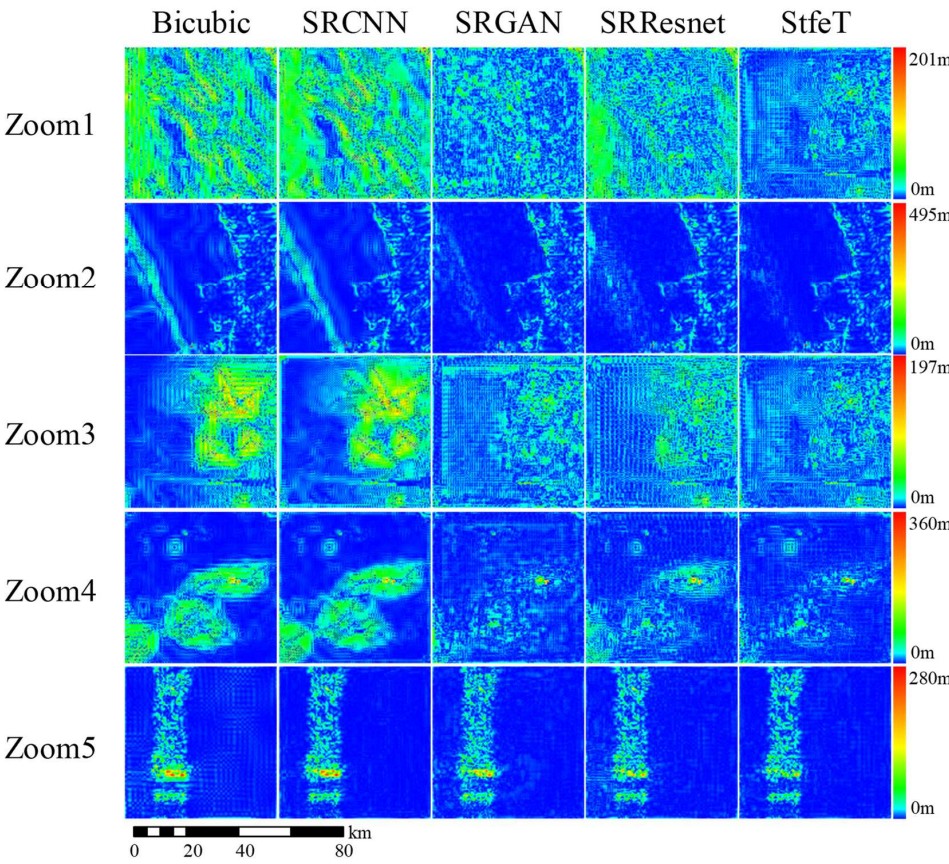

**Figure 9.** Comparison of SR DBM and HR DBM error maps for different methods in various regions.

**Table 3.** The number of errors generated using different methods.

| Method | Number (Error < 25 m) | Number (25 m < Error < 100 m) | Number (100 m < Error < 200 m | Number (Error > 200 m |
|---|---|---|---|---|
| Bicubic | 61,711 | 19,390 | 793 | 26 |
| SRCNN | 61,378 | 19,876 | 642 | 24 |
| SRGAN | 77,494 | 4344 | 60 | 22 |
| SRResNet | 71,029 | 10,704 | 163 | 24 |
| StfeT | 78,893 | 2949 | 56 | 22 |

*4.5. Ablation Study*

In this section, we verify the effectiveness of the seabed terrain feature extraction module and the efficient transformer module in DBM SR. We added the seabed terrain feature extraction module and the efficient transformer module into other models, with SRResNet as the baseline. SRResNet-dconv represents SRResNet with the seabed terrain feature extraction module, and SRResNet-ET represents SRResNet with the efficient transformer module. The comparison results are shown in Table 4.

**Table 4.** Ablation study results for the proposed STFET. Results in bold are best.

| Method | RMSE (m) | MAE (m) | Emax (m) |
|---|---|---|---|
| SRResNet | 15.13 | 7.89 | 300.07 |
| SRResNet-dconv | 15.10 | 7.70 | 322.98 |
| SRResNet-ET | 15.01 | 7.70 | 319.30 |
| STFET | **13.30** | **6.88** | **262.05** |

Common convolutional layers were used in conventional residual networks, and their convolution kernels were usually regular; however, the DBM seabed terrain features were irregular. By using deformable convolutional layers, features that more closely resembled the actual terrain could be obtained to some extent, thereby improving the accuracy of super-resolution. In addition, the efficient transformer could use a self-attention mechanism that allowed each pixel to interact with all pixels in the entire image, and this nonlocal operation could better capture long-range correlations in images, helping to recover missing details more accurately. The results in Table 3 show that the seabed terrain feature extraction module and the efficient transformer module have the potential to improve the accuracy of different deep learning-based SR methods. The effectiveness of the seabed terrain extraction module and the efficient transformer is demonstrated. However, when the seabed terrain feature extraction module and the efficient transformer module were used alone, this improvement effect was not very satisfactory. This observation shows that when using deep learning-based SR methods to extract features from DBMs, it is necessary to both extract local features and strengthen the extraction of similar global features. Only in this way can high-accuracy HR DBM be obtained more effectively.

**5. Conclusions**

In most previous studies, scholars have often used interpolation methods to improve DBM resolution, but the accuracy of generating HR DBM was poor. In this paper, we propose a seabed terrain feature extraction transformer for DBM SR, which mainly consists of the seabed terrain feature extraction module and the efficient transformer module. First, the seabed terrain feature extraction module has an important function, and it is able to adaptively extract DBM seabed terrain features. This adaptive ability enables the model to perceive DBM seabed terrain features at different scales and strengthen the extraction of local DBM seabed terrain features. Second, the efficient transformer module finds similar seabed terrain features through the self-attention mechanism, which enhances the extraction effect of similar global seabed terrain features. The effectiveness of these two modules was verified through ablation experiments, which demonstrated that, compared with bicubic interpolation and SRCNN, SRGAN, and SRResNet, STFET decreased the RMSE

by 16%, 10%, 13%, and 12%, respectively. Furthermore, especially in areas with complex seabed terrain features, the accuracy of SR DBM generated using STFET reconstruction was significantly improved.

The method proposed in this paper considers global and local seabed terrain features for DBM SR but is not limited to DBM SR. While making up for the lack of interpolation accuracy, it can also be combined with the interpolation method for data fusion. Our next step considers that the proposed STFET can be used to transfer the trained model to the sea domain with only LR DBM, and we plan to rebuild and generate HR DBMs to compensate for the deficiency of LR DBMs and meet the needs of practical applications.

**Author Contributions:** W.C. collected and processed the data, performed analysis, and wrote the Paper; Y.L. proposed the main idea and made suggestions on the experiments; Y.C. and Z.D. helped to write and edit the article; H.Y. and N.L. contributed to the validation. All authors have read and agreed to the published version of the manuscript.

**Funding:** This work was supported by the Shandong Natural Science Foundation (grant number ZR2023QD113), the Shandong Postdoctoral Innovation Project (grant number SDCX-ZG-202202041), and the Qingdao Natural Science Foundation (grant number 23-2-1-73-zyyd-jch).

**Data Availability Statement:** The digital bathymetric model data used in this paper can be downloaded at: https://www.gebco.net/data_and_products/gridded_bathymetry_data/gebco_2022/ (accessed on 10 September 2023).

**Conflicts of Interest:** The authors declare that they have no known competing financial interest or personal relationships that could have appeared to influence the work reported in this paper.

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
