# Peer review of "A Seabed Terrain Feature Extraction Transformer for the Super-Resolution of the Digital Bathymetric Model"

_remotesensing, doi:10.3390/rs15204906_

Round 1

Reviewer 1 Report

Dear Authors,

Thank you for your manuscript, remotesensing-2553646. I would like to provide my review of your manuscript in the attached PDF. Please find my comments in it. I hope it could be helpful to improve your study. Thank you.

The manuscript is well-written in English. I only found some minor errors and typos in the text and the figures. Please refer to my comments in the attached PDF for detail. Thank you.

Reviewer 2 Report

The manuscript provides an interesting approach for enhancing the analysis of bathimetry data concerning terrain feature extraction. It is well organized, the methodology is well described and the evaluation of the results retrireved by the proposed approach was done propperly. Therefore, I recommend this manuscript for publication.

Reviewer 4 Report

This paper proposed a STFET method for the super-resolution of the digital bathymetric model. The proposed method is clear and innovative. This paper is generally well written and complete and is worthy of publication. But there are still some minor problems that should be addressed carefully. 

1.      The latest literatures about the super-resolution of the digital bathymetric model can be added to show the most current relevant status of your research.

2.      A workflow should be embedded into the Methodology section to illustrate the overall process of the proposed method.

3.      “AreaX” should be modified to “ZoomX” in Figure. 9.

4.      In Section 4.4, a histogram is suggested to visualize error results.

5.      The format of the References is inconsistent. Authors should carefully refer to the writing rule of RS.

Reviewer 5 Report

A seafloor terrain feature extraction and transformation model is proposed in this paper. This model combines a seafloor terrain feature extraction module with an efficient transformation module, focusing on extracting underwater terrain features from the database. It ultimately achieves high-precision super-resolution (SR) reconstruction of DBM (Digital Bathymetric Model) data in the research area. However, the paper has several issues:

1、The paper lacks innovation as it only employs commonly used super-resolution networks to reconstruct new target data without further improvements to existing modules.

2、The experiments in this paper compare the results with older algorithms, while the paper employs a more recent Transformer-designed network, which introduces a certain degree of unfairness in the comparison.

3、The authors should explain the differences between DBMs and common image data in the paper, as well as describe the distinctions in how the network handles DBM data compared to typical super-resolution networks.

4、Although the paper uses three evaluation metrics, it does not compare them with commonly used super-resolution metrics such as PSNR, SSIM, FSIM, or no-reference metrics like ERGAS and LPIPS.

5、The paper's network structure uses ResBlock for feature extraction, but it lacks a discussion on the number of ResBlocks used. It is recommended to include relevant content to address this issue.

6、In Figure 6, the LR (Low Resolution) and HR (High Resolution) dataset presentations should be aligned vertically for better clarity.

Round 2

Reviewer 5 Report

Thanks to the author for his timely answer, and I hope that the author can further promote the super-resolution network in this unique data field. Finally, authors are advised to read the following articles carefully and make relevant citations at appropriate locations. It is hoped that articles like this will help further research related to super-resolution.

1、Y. Wang, Z. Shao, T. Lu, C. Wu and J. Wang, "Remote Sensing Image Super-Resolution via Multiscale Enhancement Network," in IEEE Geoscience and Remote Sensing Letters, vol. 20, pp. 1-5, 2023, Art no. 5000905, doi: 10.1109/LGRS.2023.3248069.

2、T. Lu, Y. Wang, J. Wang, W. Liu and Y. Zhang, "Single Image Super-Resolution via Multi-Scale Information Polymerization Network," in IEEE Signal Processing Letters, vol. 28, pp. 1305-1309, 2021, doi: 10.1109/LSP.2021.3084522.

3、Wang Y, Shao Z, Lu T, et al. A lightweight distillation CNN-transformer architecture for remote sensing image super-resolution[J]. International Journal of Digital Earth, 2023, 16(1): 3560-3579.